# Microbiota Phenotype Promotes Anastomotic Leakage in a Model of Rats with Ischemic Colon Resection

**DOI:** 10.3390/microorganisms11030680

**Published:** 2023-03-07

**Authors:** Diego Zamorano, Dinka Ivulic, Tomeu Viver, Felipe Morales, Francisco López-Kostner, Roberto M. Vidal

**Affiliations:** 1Colorrectal Unit, Clínica Las Condes, Santiago 13114, Chile; 2Programa de Microbiología y Micología, Instituto de Ciencias Biomédicas, Facultad de Medicina, Universidad de Chile, Santiago 8380453, Chile; 3Marine Microbiology Group, Department of Animal and Microbial Diversity, IMEDEA (CSIC-UIB), 07190 Esporles, Illes Balears, Spain; 4Centro de Cáncer, Clínica Universidad de los Andes, Santiago 7591047, Chile; 5Instituto Milenio de Inmunología e Inmunoterapia, Facultad de Medicina, Universidad de Chile, Santiago 8380453, Chile

**Keywords:** microbiome, anastomosis, anastomotic leak

## Abstract

Anastomotic leakage (AL) is a major cause of morbidity and mortality after colorectal surgery, but the mechanism behind this complication is still not fully understood. Despite the advances in surgical techniques and perioperative care, the complication rates have remained steady. Recently, it has been suggested that colon microbiota may be involved in the development of complications after colorectal surgery. The aim of this study was to evaluate the association of gut microbiota in the development of colorectal AL and their possible virulence strategies to better understand the phenomenon. Using 16S rRNA sequencing of samples collected on the day of surgery and the sixth day following surgery, we analyzed the changes in tissue-associated microbiota at anastomotic sites created in a model of rats with ischemic colon resection. We discovered a trend for lower microbial diversity in the AL group compared to non-leak anastomosis (NLA). There were no differences in relative abundance in the different types of microbial respiration between these groups and the high abundance of the facultative anaerobic *Gemella palaticanis* is a marker species that stands out as a distinctive feature.

## 1. Introduction

Anastomotic leak (AL) is a devastating complication of colorectal surgery that has many consequences and exposes the patient to a high risk of morbidity and mortality. Despite technological advances, AL continues to occur at an unacceptable rate in terms of procedural efficacy. 

Colorectal resection is a common surgical procedure performed on thousands of patients. The global incidence of colorectal AL varies widely in the available literature, ranging between 1 and 24% [1]. The incidence of AL after a rectum resection can be as high as 50% if asymptomatic AL diagnosed by images is included [2,3]. 

This complication has many negative implications, contributing to an increased hospital stay and a mortality rate as high as 10–16% [4,5]. To prevent the devastating consequences of AL, most surgeons perform a derivative ostomy during the resection surgery. In a study of 1066 patients who underwent rectal surgery, a derivative ostomy was performed in 70% of cases [6]. This study also showed that derivation does not prevent AL but does reduce the severity of the event. The problem with this practice, which is usually temporary, is that on a significant number of occasions, it ends up being a definitive solution [7,8].

Regarding the long-term oncological outcomes, a systematic review and meta-analysis assess the impact of AL [9]. The authors concluded that AL has a negative effect after a previous oncological resection of the rectum on local recurrence and specific survival, but not on distant recurrence. 

A recent study that analyzed the factors associated with AL concluded that there is no evidence to support poor irrigation of the anastomosis, increased anastomotic tension, different surgical techniques, and preoperative radiotherapy resulting in an alteration in the anastomosis healing [10]. Therefore, to date, the pathophysiology of this postoperative complication is not yet fully understood.

Some studies have evaluated the interaction of microorganisms in the healing of intestinal anastomosis. The original research exploring microbiota’s effects in AL was conducted in 1955 [11]. An animal (dog) model was used, and ischemic colon anastomosis was performed; the half that received enteral antibiotics showed complete anastomosis healing compared to the control group, which later developed AL, peritonitis, and death.

Although previous studies provide evidence for the hypothesis that the microbiota may have a causal role in the AL, how these pathogens colonize the gastrointestinal lumen is unknown [12]. To understand how the normal intestinal microbiota change in response to surgical intervention, a research group used sequencing of 16S rRNA to characterize changes in a rat model with partial colectomy and anastomosis [12]. This study observed an increase in the relative abundance of 500-fold and 200-fold of *Enterococcus* and *Escherichia/Shigella*, respectively, using a comparison in the anastomotic tissue on the seventh postoperative day compared to day zero. This difference was observed by analyzing the intestinal tissue after creating the anastomosis, suggesting that certain bacteria develop an adhesive phenotype, thereby facilitating certain pathogenic microorganisms to adhere to the new anastomosis. 

More recently, Shogan et al. attempt to clarify the molecular mechanism by which bacteria may be involved in AL in a rat model [13]. They demonstrated that the commensal bacterium *Enterococcus faecalis* enhanced collagen-degrading activity and also increased the ability to activate host matrix metalloprotease-9 (MMP9) in intestinal tissues through gelatinase E (*gelE*) and serine protease (*sprE*). Moreover, the authors demonstrated that in stool cultures of human patients who underwent colon resections, there is the presence of *E. faecalis* and other bacteria with collagen-degrading phenotype. In this same sense, Jacobson et al. demonstrated an additional mechanism of collagenolytic virulence by which *E. faecalis* is able to bind to and locally activate the fibrinolytic protease plasminogen (PLG). Activation depends on the binding of alpha-enolase on the surface of *E. faecalis* (and probably other as yet unidentified receptors) to PLG, leading to supraphysiological collagen degradation [14]. 

Despite these advances, the incidence of surgical complications persists, and AL is associated with a substantial risk of death and is still not fully understood. Achieving the above would require a better characterization of the microbiome and the interaction with the host to understand which bacteria are associated, how it or they act postoperatively, and how we can prevent the AL associated with microbiota changes. In this sense, to identify the bacteria that colonize ischemic tissues in AL and the possible associated virulence mechanisms, we examined the microbiota using 16S rRNA analysis in rats 6 days after surgery. As a novelty, the complete 16S rRNA gene was sequenced using PacBio, which, unlike partial 16S rRNA sequencing technologies such as Illumina, provides maximum taxonomic resolution when identifying bacteria at the species taxonomic level [15]. Thus, this study seeks to shed light on the bacteria involved in AL and their possible virulence strategies to better understand the phenomenon.

## 2. Materials and Methods

### 2.1. Animal Model

Adult Sprague Dawley male rats weighing 250–300 g were used for all experiments and were allowed food and water ad libitum. The rats were kept in a ventilated place with one rat per cage, with controlled temperature and humidity. Animals were removed from the cage only for surgery. A group of 6 rats was used to evaluate the safety and reproducibility of the technique and the same surgeon performed all the procedures. Then, the previously described anastomosis with devascularization [13] was performed on a group of 12 rats. In addition, the same 12 rats were used as controls without intervention: prior to performing the anastomosis and the opening of the gut, a segment of colon was obtained with the two ends ligated, taking care to keep this segment as airtight as possible. All animals were euthanized on the sixth postoperative day to assess the integrity of the anastomosis. 

For surgery, the rats were sedated with 40–80 mg/kg ketamine and 5–10 mg/kg intraperitoneal xylazine and operated on using an aseptic technique. A laparotomy was performed, followed by a 0.5 cm distal colon resection with a rectosigmoid termino-terminal anastomosis using 12 stitches of non-absorbable 6–0 interrupted monofilament. The distal resection area was located 1 cm from the peritoneal reflection. The adjacent mesentery was divided 1 cm above and 1 cm below the anastomosis to perform anastomosis devascularization. The airtightness was then verified using a physiological serum test. 

To evaluate non-leak anastomosis (NLA) during the autopsy, it was considered that the anastomosis had to be completely intact, virtually free of adhesions, free of stenosis or inflammation (mild and compressible), and free of proximal intestine dilatation. On the other hand, those with anastomotic leak (AL) had to meet certain criteria: (1) visible adhesions to the anastomotic line, including adjacent organs; (2) compression with a cotton swab showed the anastomosis felt hard, inflamed, and incompressible; (3) dissection of the anastomotic site in the presence of a plastron with adjacent organs revealed an evident disruption or leakage of purulent content; and (4) opening of the anastomosis showed ulceration or luminal disruption. The protocol was approved by the Ethics Committee of Clínica Las Condes, Santiago, Chile.

### 2.2. Sample Processing

Total DNA was extracted from biopsy samples using the EZNA DNA/RNA Isolation kit (Omega-Bio-Tek, Norcross GA, USA) following the manufacturer’s recommendations. One extracted aliquot per sample was shipped to MR DNA (Molecular Research LP, Shallowater, TX, USA) for 16S rRNA gene library construction and sequencing.

### 2.3. Primers, 16S rRNA Amplification, and Sequencing Procedure

Primers GM3 (5′-AGAGTTTGATCMTGGC-3′) and 23S1 (5′-GGGTTTCCCCATTCGGAAATC-3′) with a unique barcode linked to the forward primer were used in the amplification of the target sequence. These primers comprise the complete 16S rRNA gene plus the contiguous internal transcribed spacer (ITS) located between 16S and 23S rRNA genes, a region with a high degree of variability, allowing for additional analysis on relatedness and discrimination of intra-species diversity [16]. A 35-cycle PCR (5 for PCR products) was performed using the HotStarTaq Plus Master Mix Kit (Qiagen, MD, USA). PCR settings were 94 °C for 3 min, then 35 cycles of 94 °C for 30 s, 53 °C for 40 s, and 72 °C for 90 s, with a final elongation step of 72 °C for 5 min. The amplification performance was inspected in 2% agarose gel to examine the relative intensities of the PCR products. Next, various samples were joined together in equal parts based on their molecular weight and DNA concentrations. The PCR pool was purified using Ampure PB beads (Pacific Biosciences, Menlo Park, CA, USA). The SMRTbell library preparation was accomplished using the SMRTbell^®^ Express Template Prep Kit 2.0 following the user guide. Finally, the sequencing process was completed at MR DNA (www.mrdnalab.com accessed on 30 January 2022, Shallowater, TX, USA) on the PacBio Sequel platform following the manufacturer’s guidelines.

### 2.4. Processing and OTU (Operational Taxonomic Units) Grouping

After finalizing the initial DNA sequencing, the library underwent a second analysis, Circular Consensus Sequencing (CCS), using PacBio’s CCS algorithm (SMRT Link 9.0.0.92188). The CCS algorithm aligns the subreads individually from each amplicon to generate consensus sequences, allowing the correction of stochastic errors caused in the first analysis. Sequence data were processed using the MR DNA analysis pipeline (MR DNA, Shallowater, TX, USA). In short, barcodes were separated from the CCS sequencing data, then oriented 5′ to 3′, and sequences < 150 bp were removed. Chimera removal and de novo OTU clustering were carried out using the USEARCH v.7 software package. Operational taxonomic units were defined by grouping sequences at a similarity threshold of 99%.

### 2.5. Phylogenetic Affiliation and OPU (Operational Phylogenetic Unit) Clustering

OTU representative sequences were included in the nonredundant Silva 138 database using the ARB program package [17,18]. Sequences were aligned using SINA (SILVA Incremental Aligner) [19] and then added to a preexisting tree to choose the closest relatives and generate a phylogenetic tree reconstructed by neighbor-joining. The tree was modified using the parsimony tool to optimize the branching order. Finally, the parsimony tree was manually curated to obtain OPUs.

An OPU is the smallest monophyletic clade clustering OTU representatives and their closest reference sequence. Where feasible, OPUs included a type strain as a reference sequence. When this was not possible, other available databases were consulted. For identity values >98.7% containing a type strain sequence, reads were considered as belonging to the same species. When identity values were <98.7 and >94.5% with the closest relative type strain, reads were different unclassified species of the same genus [20]. This focus uses phylogenetic inference rather than clustering by sequence identity to show diversity measures as group sequences in lineages that approach the species thresholds with a more robust view of microbial diversity [21].

### 2.6. Alpha Diversity, Beta Diversity, Comparative, and Statistical Analyses

Alpha diversity indices and rarefaction curves were calculated using the PAST software v4.05 [22] on bacterial reads of each sample. For data analysis, OPUs harboring <2 sequences and those occurring in a single sample were removed for beta diversity analysis and comparative analysis. Data scaling was performed by the total sum scaling (TSS) method, transforming raw reads into relative abundances by dividing the number of reads of an OPU in each sample by the total number of reads. The MicrobiomeAnalyst web-based tool [23] was used to conduct the OPU profile analyses at different taxonomic levels and the β-diversity analyses (PCoA, HeatTree, LEfSe, and correlations network). The options and parameters selected can be found in the description of each figure. The annotation of microbial respiration was generated manually, using information from SILVA references [18] and papers related to the description of each species). Venn diagrams were carried out with the “InteractiVenn” web application [24]. Statistical differences between groups were performed utilizing the stats v4.1.2 R package [25]. Normality and homoscedasticity were checked. Then, parametric or non-parametric tests were performed. For post hoc pairwise comparisons, a Bonferroni- or Tukey-adjusted test was performed. 

## 3. Results

### 3.1. Samples, Sequences, and OPUs Distribution

A total of 12 rats underwent surgery and 24 samples were analyzed, with 12 being control mucosa samples (M) and 12 being experimental anastomosis samples (A). Of these, 4 were classified as NLA and 8 as AL (Figure 1). A process control sample was also included (without a piece of tissue). A total of 349,633 reads were obtained that grouped into OTUs at 99% sequence identity, yielding 2484 different OTUs, with an average of 201 (±154) OTUs for control samples and 145 (±68) OTUs for anastomosis samples. Among the samples of the anastomosis group, a total of 175 (±115) OTUs were obtained for the NLA samples and 130 (±27) OTUs for the AL samples. Finally, a total of 317 different OPUs were generated by phylogenetic inference using OTU representatives, with an average of 39 (±28) OPUs for control samples and 53 (±29) OPUs for experimental samples. Among the samples of the anastomosis group, a total of 70 (47±) OPUs were obtained for the NLA samples and 45 (±12) OPUs for the AL samples (Appendix A).

After removing OPUs with identical values (zeroes) in all samples, as well as OPUs appearing in only one sample (considered artifacts), the data set was reduced from 317 to 169 total OPUs (Appendix A). Of these, 106 OPUs (62.7%) were shared between control and anastomosis samples, 18 OPUs (10.7%) were present exclusively in control samples, and 45 OPUs (26.6%) were present only in anastomosis samples (Appendix A). When analyzing samples within the anastomosis group, 105 OPUs (69.5%) were shared between NLA and AL samples, 30 OPUs (19.9%) were present exclusively in NLA samples, and 16 OPUs (10.6%) were present only in AL samples (Appendix A). This accounts for a lower diversity of OPUs in the AL group than in the NLA group.

### 3.2. Alpha Diversity Involvement in Anastomotic Leak

Diversity represents the number of species and their distribution, described in terms of alpha diversity as the different species found within a sample. This diversity is believed to contribute to greater stability of an individual’s microbiome [26], which may protect against external-induced change. Accordingly, we analyzed the alpha diversity and its variation in the animals that underwent surgery.

In Figure 2A,B, rarefaction curves are shown for the number of OTUs, OPUs, and reads analyzed. The OTU approach (Figure 2A) showed rarefaction curves that saturated later than the OPU rarefaction curves (Figure 2B), indicating that the traditional OTU approach may overestimate diversity [20]. Both the OTU (Figure 2A) and OPU (Figure 2B) graphs show that the samples reach the sequencing plateau, though the OPU approach does so earlier. It is noted that the sequencing depth was optimal, particularly when analyzing the rarefaction curves of the OPU approach, indicating that the greatest diversity of samples was recovered. It should be noted that the values of Chao-1, which represents the number of estimated OPUs (Appendix A), and the number of observed OPUs (Appendix A) for all samples analyzed coincide, indicating again the optimal sequencing depth obtained for all libraries.

When the control (M) and experimental anastomosis (A) groups were analyzed, it was found that they both had similar alpha diversity indices, without showing significant differences. The same happens when analyzing the NLA and AL groups (Figure 2C, Appendix A). Even when comparing the three groups, M, NLA, and AL, no differences were observed. Despite this, a trend towards a lower presence of estimated OPUs in Chao-1 can be observed for the AL group when compared to the NLA group. In addition to this, there is also a trend towards greater dominance and less heterogeneity in AL given the Shannon–Weiner Index. Finally, lower values for the AL group can be seen in evenness, indicating a less uniform distribution of species (OPUs). All these results show us a tendency for less diversity in the AL samples.

These results are in line with what was previously described, where low diversity has been observed in the microbiota of the anastomotic tissue that develops AL [27,28,29]. This may be so because reduced diversity may be associated with an increase in undesirable microbiota and a decrease in beneficial microbiota, resulting in compromised intestinal resistance [30]. It is well known that the microbiota serves the host in different ways, such as the competitive exclusion of pathogens or digesting complex carbohydrates to produce the SCFAs (short-chain fatty acids) needed as energy sources for the enterocyte [31,32], among others. Thus, in the same way, in inflammatory bowel disease, reduced bacterial diversity is described [33], which affects the ability of the intestinal microbiome to maintain a normal profile thanks to the services provided by the bacterial community.

### 3.3. Changes in the Intestinal Microbiota Associated with Surgical Anastomosis

To explore the change of structure in the mucosal microbiota of rats undergoing anastomosis compared to samples obtained from unoperated control rats, a beta diversity analysis was performed. For this, PCoA analyses were used to compare the microbiota of M (control) and A (anastomosis) (Figure 3). There were significant differences in the composition of these groups, with the two main PCoA axes explaining 46.5% of the variation (PERMANOVA, *p* = 0.001). A significant clustering pattern (28.5%) was found on axis 1, where the statistical test revealed two distinct microbiome profiles. A significant clustering pattern (18%) was also observed on axis 2, where two different microbiome profiles between A and M were observed.

Then, a HeatTree was performed to compare the differences between samples A and M in order to observe distinctive features at each taxonomic level. Thus, the differences revealed by the beta diversity analysis could be explained by differences observed in Figure 4, which illustrates differences between the two groups. Here, 4 phyla, 5 classes, 8 orders, 16 families, 21 genera, and 33 species (Appendix A) were statistically different. At the phylum level, *Firmicutes*, *Proteobacteria,* and *Fusobacteriota* stand out as being more represented in A than in M. Within *Firmicutes*, the *Bacilli* class stands out, including the species *Gemella palaticanis*. In the phylum *Proteobacteria*, the class *Gammaproteobacteria*, the order *Enterobacterales,* and the family *Pasteurellaceae* within which the genus *Rodentibacter* stands out. In the *Bacteroidaceae* family, the genus *Bacteroides* stands out. In the phylum *Fusobacteriota* the species *Fusobacterium gastrosuis* stands out. By contrast, the phylum *Campylobacterota* stands out as being more represented in M than in A. In this phylum, the genus *Helicobacter* stands out.

### 3.4. Bacterial Biomarkers of Anastomotic Intervention

Then, a univariate “LEfSe” analysis was performed to identify microbial species that differed significantly between groups A and M. LEfSe first employs the Kruskal–Wallis test to identify taxa with significantly different relative abundances between groups [23]. Using a raw *p*-value cutoff of 0.05, 33 species were identified as distinct, which coincided with the 33 species identified as different in the comparative heat tree A vs. M (Figure 4, Appendix A) using the Wilcoxon test. The LDA (linear discriminant analysis) method was then used to generate a ranked species list based on their LDA scores. The above enabled us to identify the taxa that best represent each group (Figure 5A). As a result, the anastomosis group presented 28 marker species. This group was significantly (p-adjusted) dominated by five species: *Gemella palaticanis, Rodentibacter ratti, Bacteroides sartorii, Fusobacterium gastrosuis, and Bacteroides uniformis*. By contrast, five species were found to be marker species of the mucosal control group, none of which were statistically significant (p-adjusted) (Figure 5A, Appendix A).

A correlation network was built (Figure 5B, Appendix A) over the 33 LDA LEfSe species to uncover potential bacterial interactions that could aid in understanding how the microbiota responds and adapts to anastomosis intervention. Notably, *Gemella palaticanis* and *Rodentibacter ratti* showed by far the highest positive correlation (0.8373). Then, the highest negative correlation was between *Gemella palaticanis* and *Helicobacter muridarum* (−0.5158). The second highest negative correlation was observed between *Bacteriodes uniformis* and *Helicobacter muridarum* (−0.5107). It is noteworthy that the highest positive correlations were observed between *Gemella palaticanis* and *Rodentibacter ratti*, both of which are facultative anaerobes and had the highest LDA scores within the anastomosis group. It is also noteworthy that the highest negative correlation values pointed to the microaerophilic *Helicobacter muridarum*, which is among the marker species of the control mucosa group. As a result of these observations, the next focus was on investigating what type of microbial respiration of the intestinal microbiota characterized each group.

### 3.5. Changes in Microbial Respiration of the Intestinal Microbiota

The healthy intestinal mucosal environment is characterized by a low oxygen level that allows the survival and establishment of anaerobic bacterial communities. By contrast, during anastomotic surgery, the intestinal mucosa is exposed to high concentrations of oxygen that can affect and stop the growth of obligate anaerobic bacteria [34]. On this basis, the microbial respiration of mucosa-associated intestinal microbiota was investigated in the anastomosis and control groups. It was found that the anastomosis group (A) had a higher abundance of facultative anaerobes and a lower abundance of microaerophiles (*p* < 0.005). A higher abundance of aerobes was also observed in the mucosal control group (M). No differences were observed in the abundance of the obligate anaerobes (Figure 6A,B, Appendix A).

As a result of these findings, oxygen appears to play an important role in the intestinal shift of the anastomosis phenotype. This is consistent with what has been previously suggested, where oxygen exposure is among the factors that could be implicated in changes in the composition of the intestinal microbiota due to colorectal surgery [35]. Prior evidence [12,36] showed that exposure of the intestinal mucosa to oxygen is associated with a loss of obligate anaerobes and a gain in potentially pathogenic facultative anaerobes such as *Enterococcus and Escherichia/Shigella*. In the present study, no changes were observed in the relative abundances of obligate anaerobic bacteria, but a significant increase in facultative anaerobes was indeed observed, mainly at the expense of a decrease in the relative abundance of microaerophiles when comparing the anastomosis group with the control group. As can be seen in Figure 5A, these changes are mainly due to the increase in the abundance of the facultative anaerobes *Gemella palaticanis* and *Rodentibacter ratti*, and the decrease in the microaerophiles *Helicobacter* sp. (OPU 361) and *Helicobacter muridarum*. This can be observed more globally in Figure 7, where the total absence of *Gemella palaticanis* and the minimal presence of *Rodentibacter ratti* in the mucosal control group stand out compared to the anastomosis group.

### 3.6. Microbiota Leak Phenotype

This study describes a trend for lower microbial diversity in the AL group compared to NLA (Figure 2C). There were also no differences in relative abundance in the different types of microbial respiration between the NLA and AL groups (Figure 6C). Next, the differences in the structure of microbial communities between NLA and LA were explored, as well as species potentially involved in the pathogenesis of an anastomotic leak.

To explore the differences in the structure of microbial communities between NLA and LA samples, a beta diversity analysis was performed. For this, PCoA analyses were used to compare the microbiota of NLA and AL (Figure 8A). No differences were observed between these two groups. This shows that globally, the two microbial communities did not differ. Next, it was studied whether there were specific species (OPUs) that could account for differences between the two groups. For this, a LEfSe analysis with LDA was performed using a raw *p* cutoff value of 0.05. The results made it possible to identify 11 distinct species, with 9 marker species for the NLA phenotype and 2 marker species for the AL phenotype (Figure 8B, Appendix A). Thus, *Gemella palaticanis* and *Lactobacillus* sp. (OPU 113) were the species that best represented the AL phenotype. On the other hand, the species that best represented the NLA phenotype were *Helicobacter rodentium*, *Lactobacillus murinus*, *Ligilactobacillus* sp. (OPU 116), *Lactobacillus reuteri*, *Rhodospirillales* sp. (OPU 355), UCG_005 sp. (OPU 31), *Clostridium sensu stricto* sp. (OPU 145), *Muribaculaceae* sp. (OPU 199), *Muribaculaceae* sp. (OPU 209).

Then, a correlation network (Figure 8C, Appendix A) was built between all NLA and AL species to plot only those present in the LEfSe-LDA of Figure 8B. This was performed to discover possible bacterial interactions that could help in understanding how the anastomotic leak phenotype (AL) behaves in relation to the non-leak phenotype (NLA). Here, a strong positive correlation (>0.9) was observed between the obligate anaerobes *Rhodospirillales* sp. (OPU 355), *Muribaculaceae* sp. (OPU 199), and UCG_005 sp. (OPU 31), which correspond to marker species in the NLA group. A strong positive correlation (0.936) was also observed between *Clostridium sensu stricto* sp. (OPU 145) and *Muribaculaceae* sp. (OPU 209), both obligate anaerobes that also correspond to marker species in the NLA group (Figure 8B,C). This marked presence of obligate anaerobes may be illustrative of their importance in maintaining a NLA phenotype.

Next, a negative correlation (−0.601) was observed between the facultative anaerobe *Gemella palaticanis* and the microaerophile *Helicobacter rodentium*. Both were the most important marker species of the AL phenotype (LDA score −6.13) and the NLA phenotype (LDA score 4.58), respectively (Figure 8B,C). This behavior is similar to what was seen between the anastomosis group (A) and the control mucosal group (M), where the most important marker species for each group corresponded to facultative anaerobes and microaerophiles, respectively (Figure 5A). Thus, the possible effect due to oxygen exposure during the anastomosis procedure seems to play an important role not only in the change in the intestinal microbiota of the anastomosis/mucosal control groups, but also in the NLA/LA phenotypes. In both cases, the high abundance of the facultative anaerobe *Gemella palaticanis* as a marker species stands out as a distinctive feature. This was manifested as a null presence of this bacterium in the control group compared to the anastomosis group, and as a more marked presence in the AL phenotype, with a relative abundance of 35.6% (±25.8) compared to the NLA phenotype, the relative abundance of which was 8.5% (±8.3) (Figure 7).

*Lactobacillus murinus* and *Ligilactobacillus* sp. (OPU 116), both facultative anaerobes and marker species of the NLA phenotype, showed a positive correlation (0.641) (Figure 8B,C). Thus, the presence of an anaerobic facultative marker species is not exclusive to the AL phenotype. Finally, *Lactobacillus reuteri* and *Lactobacillus* sp. (OPU 113) did not show any correlation with the rest of the marker species. *Lactobacillus reuteri* is another anaerobic obligate species, thus 6 of 9 species corresponded to this type of microbial respiration in the NLA phenotype. *Lactobacillus* sp. (OPU 113), for its part, is a microaerophile or aerotolerant anaerobe, a marker species of the AL phenotype together with *Gemella palaticanis*, and which demonstrates that microaerophiles are not exclusive marker species of the NLA group (Figure 8B,C).

## 4. Discussion

The best animal model [37] should be able to reproduce the disease or event under study with the greatest anatomical and physiological resemblance to a human. However, it must also be widely available, inexpensive, and technically feasible to perform in a safe and repeatable manner, and the animal should be easy to keep in a suitable space.

Animal models used are usually very different from the typical clinical setting in humans. Models are generally young and healthy animals, whereas humans are typically old and fragile. The above would explain why AL is less common in animal models, so some pathophysiological mechanisms (e.g., insufficient sutures, ischemia) must be simulated for filtration to occur [38,39].

AL models have been performed in rats and mice. These animals are excellent for these purposes because they are low-cost, easy to maintain and handle, and widely available. Unlike strict herbivores, these animals can digest food derived from animals, and their digestive system is less fragile, being a model of greater similarity to humans [40]. Most studies have been conducted in rats with discouraging initial results, where despite models with insufficient or ischemic sutures, the filtration could not be reproduced. The authors conclude that due to the greater consistency of the stools, creating a colorectal anastomosis that presents a leak is harder [41]. In other models with the use of corticosteroids, AL was evidenced, but the omentum contained it without showing an obvious filtration [42]. Recently, rats with ischemic colorectal anastomosis were used, which was achieved by the devascularization of a proximal and distal centimeter of the anastomosis. A filtration rate of 50% confirmed by the autopsy of the specimens on the sixth day was achieved postoperatively, showing only one death during anesthesia [43], which makes it an excellent model to study this event, being reproducible, economical, with a high incidence of the event to be studied and safe [37].

Like the Shogan study [13], our rat model has certain limitations in terms of recapitulating the anastomotic leak in humans. Leaks are not directly visualized during surgery as most patients do not undergo reoperation. In addition, most anastomotic leaks are discovered by computed tomography (CT) scans and appear as inflammation or fluid collections adjacent to the anastomotic site. The precise definition of AL is thus debated, and contrast enema and CT scanning are known to carry significant false negative rates. Our approach examines all anastomoses in rats for evidence of dense adhesion, dehiscence, inflammation, and purulence that contrast with a healthy non-inflamed anastomosis without adhesions. We believe this assessment aligns with what might be seen clinically in patients.

The gut microbiota has been proposed as a critical factor in colorectal surgery complications. It has been hypothesized as the “missing factor” that could explain little-understood complications of an anastomotic leak [44]. Bacterial diversity (alpha diversity) is believed to contribute to the stability of the microbiome. It was previously reported that there are postoperative changes in the microbiome after gastrointestinal surgery, where better outcomes are associated with greater microbiome diversity [26]. It has also been reported that low microbial diversity may decrease resistance to colonization by pathogenic bacteria that might play a role in AL development [27,28]. Although our results showed no differences between the anastomosis group and the mucosal control group, we found a trend toward lower microbial diversity in the AL group compared to the NLA group.

The current study shows that the intestinal anastomosis procedure induces changes in microbial composition after surgery in rats. Other factors, such as sex, age, diet, or the animal model’s comorbidities, could influence how the microbiota affects AL but was not considered in this study. However, the effect of oxygen on bacterial microbiota is a general mechanism that is sufficiently independent of host factors (i.e., gender) to account for our findings, which can also be seen in Chamorro’s findings [21] regarding the effect of oxygen on the microbiota of female and male IBD patients. Thus, we observed significant differences in the composition (beta diversity) between the anastomotic intervention group and the mucosal control group. A possible explanation for this change in the structure of the communities may be the exposure to a high concentration of oxygen when the intestine is opened during anastomotic surgery [35]. Our results support this idea, showing a marked increase in the abundance of anaerobic, facultative bacteria in the anastomotic group at the expense of a decrease in microaerophiles. This change was mainly due to the high presence of the facultative anaerobic bacterium *Gemella palaticanis* in the anastomosis group.

Previous studies have described a marked increase in the abundance of anaerobic, facultative bacteria at the anastomosis site. Thus, Shogan et al. showed a 500- and 200-fold increase in the relative abundance of *Enterococcus* and *Escherichia/Shigella*, respectively [12,13]. In addition, these enterococci showed the ability to degrade collagen and activate the production of matrix metalloproteinase 9 in the host, thanks to the virulence factors *gelE* and *sprE* of *E. faecalis*. Consequently, it is likely to weaken the anastomosis by degrading collagen in the healing tissue [13,45].

Furthermore, Lam et al. [46] mentioned that rodent models fed a high-fat and low-fiber diet induced alterations in the gut microbiota composition that might influence the postoperative process in anastomotic leakage; however, it showed an attenuation based on a short course of a healthy high-fiber diet before surgery [46,47]. The role of microbiota (“invisible enemy“) in the AL phenotype must be uncovered, as has been mentioned by Lam et al. [46] to prevent strains with adhesive phenotype and tropism towards injured tissues. In our study, neither control nor surgical rats had *Enterococcus* in their colorectal mucosa microbiota. Shogan’s work and our study both characterized the mucosa-associated microbiota; however, we used different strains of rats (Sprague Dawley), which could explain the differences in results, as Li et al. suggest [48]. Li mentioned that differences in gut microbiota might account for the animals’ differential metabolic response to dietary intervention and, as a result, predispose them to different pathological outcomes.

On the other hand, the PacBio platform was used in our work to characterize the microbiome, and the complete sequence of the 16S RNA plus the ITS region was obtained. This undoubtedly contributes to greater specificity in the phylogenetic association of the OPUs discovered [49]. Still, we may have lost sensitivity to detect *Enterococcus*, which is also associated with its low presence in the early microbiota.

This study observed no differences in composition (beta diversity) between the NLA and AL phenotypes. There were also no differences in the types of microbial respiration between the two groups. However, the presence of obligate anaerobes was observed, as marker species (LEfSe-LDA) in the NLA phenotype and not in AL. We also identified that these anaerobic obligate species had a strong positive correlation with each other. The importance of anaerobic obligate bacteria in maintaining the stability of the gut microbiota has previously been reported [21,34]. These obligate anaerobes are very important to maintain the environmental stability of the intestinal tract and offer services to the host, such as SCFA production [32]. For example, SCFA butyrate has been shown to have anti-inflammatory effects and is also a primary source of energy for colonic epithelium [50]. Thus, our results show that the presence of obligate anaerobic marker species related to the NLA group (*Lactobacillus reuteri*, *Rhodospirillales* sp. (OPU 355), UCG_005 sp. (OPU 31), *Clostridium sensu stricto* sp. (OPU 145), *Muribaculaceae* sp. (OPU 199), *Muribaculaceae* sp. (OPU 209), Figure 8B) may be playing an essential role in maintaining the non-leakage anastomosis phenotype.

Additionally, the presence of facultative anaerobic bacteria as a marker (LEfSe-LDA) was observed in both the AL and NLA groups. Among these, the AL phenotype’s most prominent marker species was *G. palaticanis* [51]. This bacterium showed a relative abundance of up to 80.4% in the AL phenotype samples (Figure 7). Therefore, our results show that *G. palaticanis* may play an important role in maintaining the AL phenotype when its population increased during the surgery. Thus, although the compositional structure and the type of respiration of the NLA and AL phenotypes do not differ, there are bacterial marker species that would be involved in the manifestation of the anastomotic leakage phenotype and affect wound healing.

*Gemella* species is a Gram-positive coccus, catalase-negative, facultatively anaerobic bacterium in human mucous membranes, human dental plaque, oral cavity (oropharynx), the genitourinary system, upper respiratory tract, and gastrointestinal tract [52,53]. There are six bacterial species, *Gemella hemolysins, G. morbillorum, G. bergeri, G. sanguinis, G. asaccharolytica,* and two species recovered only from animals, *G. cuniculi* and *G. palaticanis* [54]. In addition, *Gemella* has been associated as a causative agent in different human infections, such as bacteremia, age-related macular degeneration (AMD), the microbiome in cancerous tissue where *Gemella* was observed in stages I, II, and IV, root caries, and rarely in infective endocarditis [55,56,57,58]. In addition to the preceding, Liu et al. [59] describe a three-level relationship between intestinal microbiota and anastomotic leakage (AL). The first are those associated with peri- or intraoperative factors, the second with how pathogenic bacteria adhere to the intestinal mucosa and interact with the extracellular cell matrix (ECM) to prevent host clearance, and the third with how pathogens affect anastomotic healing.

The analysis of the *G. palaticanis* reference genome (ASM1523476v1) and protein table, which were deposited and annotated in GenBank, enabled us to identify some virulence factors that could explain the rise of this opportunistic pathogen in rats with AL. Fibronectin-binding protein A (FnBPA), a cell surface-bound protein that binds fibronectin and fibrinogen, and the SdrC-like adhesin that promotes cell attachment and aggregation by an unknown mechanism, are found in *G. palaticanis* [60,61]. Furthermore, the genome of *G. palaticanis* (Accession number NZ JADGKN010000100.1) harbors a putative holin-like toxin that may oligomerize in the cytoplasmic membrane and trigger the formation of holes that cause cell death, different families of serine proteases (S1C, ClpP, ClpX), a type II and III toxin-antitoxin system (RelE/ParE, RelB/DinJ, RelE/StbE, PemK/MazF, ToxN/AbiQ), metallo-endopeptidase, metallo-hydrolases and M28, Xaa-Pro and U32 family of peptidases, among other putative virulence genes.

Authors should discuss the results and how they can be interpreted from the perspective of previous studies and of the working hypotheses. The findings and their implications should be discussed in the broadest context possible. Future research directions may also be highlighted.

## 5. Conclusions

To summarize, *G. palaticanis* and *E. faecalis* can both colonize various niches in their hosts, including the oral cavity [55,62,63,64,65], the heart [66,67,68,69], and the gut [70,71,72,73]. Moreover, we describe *G. palaticanis* as a new bacterium that may be involved in generating the AL phenotype in our murine model. Our findings suggest that when the intestine is opened, oxygen promotes the growth of this bacterium (among other facultative anaerobes), which then finds a favorable niche for adhesion where the extracellular matrix at the site of anastomotic injury is exposed. This occurs in both the NLA and AL groups, but the presence of *G. palaticanis* is more significant in the AL phenotype. In this sense, preoperative stool sampling to detect patients at high risk of developing AL appears to be ineffective currently, as recent data are too heterogeneous to identify a harmful bacterial composition of the bowel. Instead, it seems that AL is caused by a combination of factors and is not due to surgery alone. On the contrary, positive associations with other facultative anaerobes could indicate cooperative or symbiotic interactions in which the AL phenotype is a consequence of a dysregulation in the proportions of these populations due to oxygen influx and the possibilities they have to colonize surgery-associated tissue.

## Figures and Tables

**Figure 1 microorganisms-11-00680-f001:**
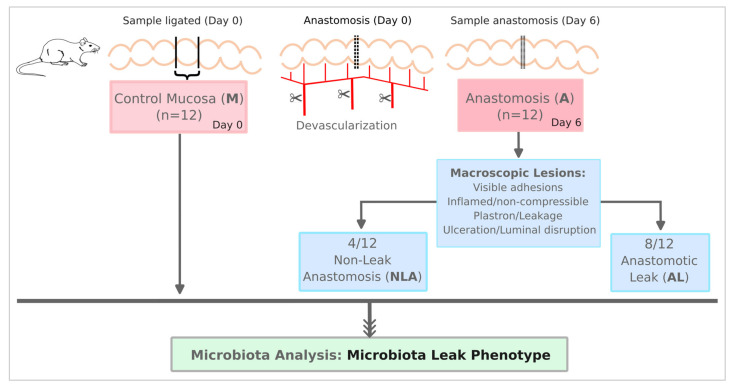
Grouping scheme of rats subjected to anastomosis and controls.

**Figure 2 microorganisms-11-00680-f002:**
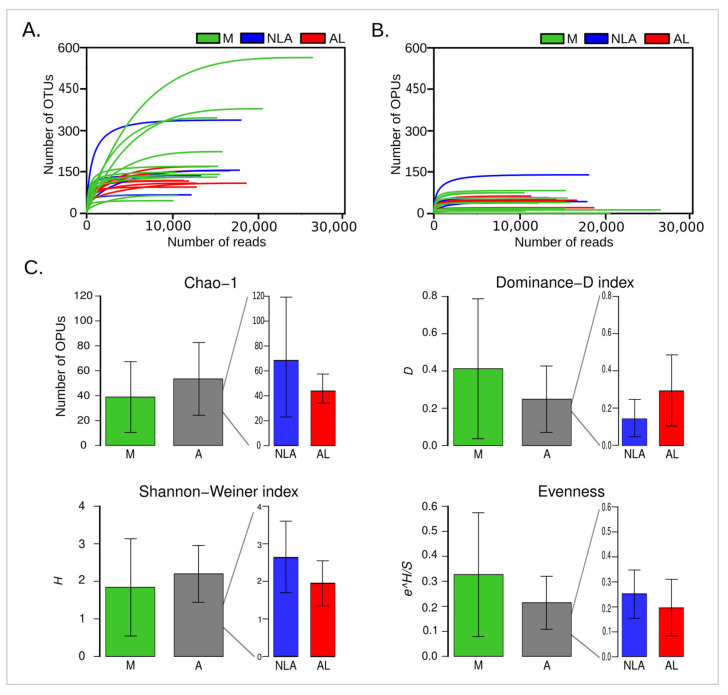
**Rarefaction and alpha diversity.** Rarefaction curves for detected OTUs (**A**) and OPUs (**B**), where each line represents an independent sample. (**C**) Alpha diversity indices, where the average diversity index ± standard deviation for each group is plotted. The Wilcoxon rank sum test was used to compare differences (* *p* < 0.05) between groups. M: control mucosa group, A: experimental anastomosis group, NLA: non-leak anastomosis, AL: anastomotic leak.

**Figure 3 microorganisms-11-00680-f003:**
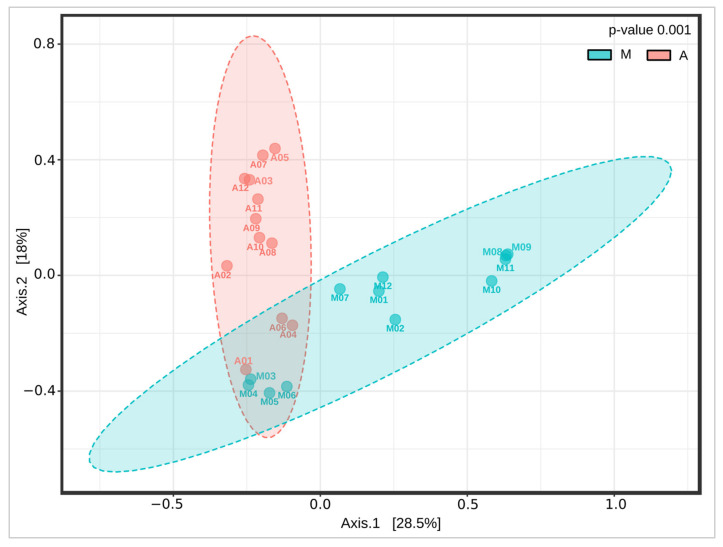
**PCoA plot for beta diversity analysis.** Samples A4, A6, A8, and A9 correspond to NLA. The remaining A samples correspond to AL. The analysis was performed using Bray–Curtis measures of beta diversity. Statistics: PERMANOVA. A: anastomosis, M: control. F-value 4.8374; R-squared 0.18025; *p*-value 0.001.

**Figure 4 microorganisms-11-00680-f004:**
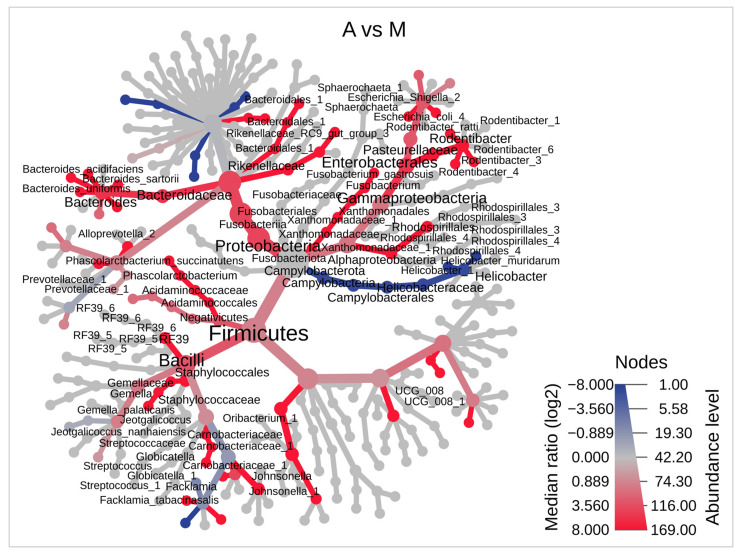
**Taxonomic differences are visualized using a heat tree.** The heat tree depicts the taxonomic differences between the experimental anastomosis (A) and control (M) groups, up to the species level. On the heat tree, only significant taxon names are labeled. Colors indicate whether corresponding taxa are lower (blue) or higher (red) in experimental rats compared to controls. The median abundance and the statistical non-parametric Wilcoxon rank sum test (*p* < 0.05) were used.

**Figure 5 microorganisms-11-00680-f005:**
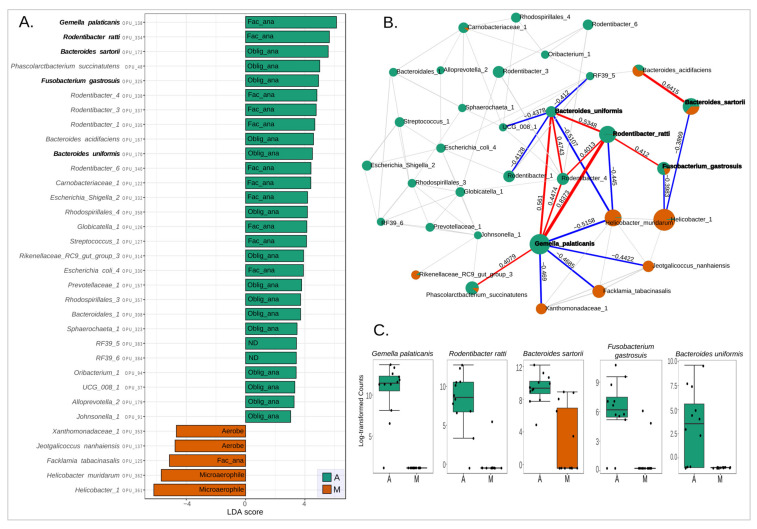
**Anastomosis group and control group species abundance and correlation network analysis.** (**A**) LEfSe-LDA score >2 identified species that differ significantly in abundance between groups (raw *p*-value cutoff < 0.05, Kruskal–Wallis test). Species with a Kruskal–Wallis FDR-adjusted *p*-value < 0.05 were highlighted. (**B**) Correlation networks (SparCC) of 33 species depicted in LEfSe-LDA. Correlation threshold > 0.3 and *p*-value threshold < 0.05 were used. The nodes (pie charts) represent species, size reflects relative abundance, with the anastomosis group in green and the control group in orange. Correlation values of species with an FDR-adjusted *p*-value < 0.05 in LDA-LEfSe were labeled, along with their connections. Positive and negative correlations in red and blue lines, respectively. The thickness of the line represents the strength of the correlation. (**C**) Boxplot of the abundances of species with an FDR-adjusted *p*-value < 0.05 in LDA-LEfSe. Fac_ana: facultative anaerobe, Oblig_ana: obligate anaerobe, ND: unknown respiration type, M: control mucosa, A: anastomosis.

**Figure 6 microorganisms-11-00680-f006:**
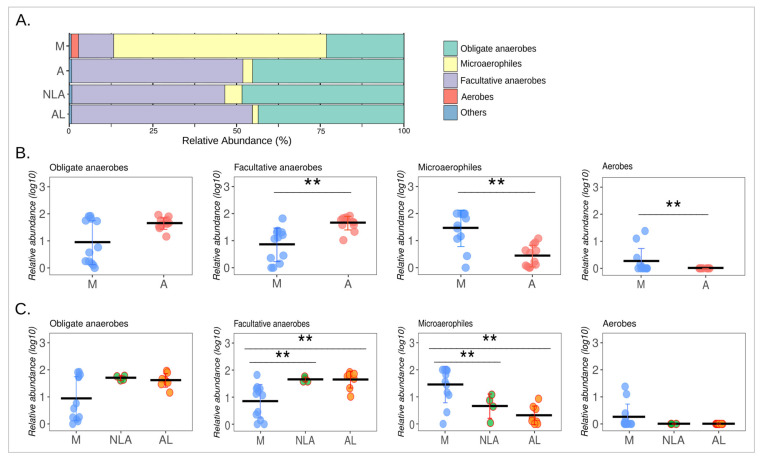
**Microbial respiration of microbiota.** (**A**) Relative abundances (%) of species classified by respiration type. (**B**) For M and A comparations, Wilcoxon rank sum test was used. (**C**) For M, NLA and AL, Kruskal–Wallis and Bonferroni’s post hoc were performed. (**B**,**C**) graphs depict relative abundances (Log10) where each data point represents a sample, and the horizontal lines represent the means. Error bars depict standard deviations. Statistical significance: ** *p* < 0.005. M: control mucosa, A: anastomosis, NLA: non-leak anastomosis, AL: anastomotic leak. The same analysis was then performed but dividing the anastomosis group (A) into non-leak anastomosis (NLA) and anastomotic leak (AL) to compare all the groups. Here, it was observed that NLA and AL, like the anastomotic group (A), showed a higher abundance of facultative anaerobes (*p* < 0.005), a lower abundance of microaerophiles (*p* < 0.005), with no differences in obligate anaerobes compared to the control group. No differences were observed between groups in the category of aerobes. Furthermore, no difference was observed between NLA and AL in any of the categories of microbial respiration (**A**,**C**, Appendix A).

**Figure 7 microorganisms-11-00680-f007:**
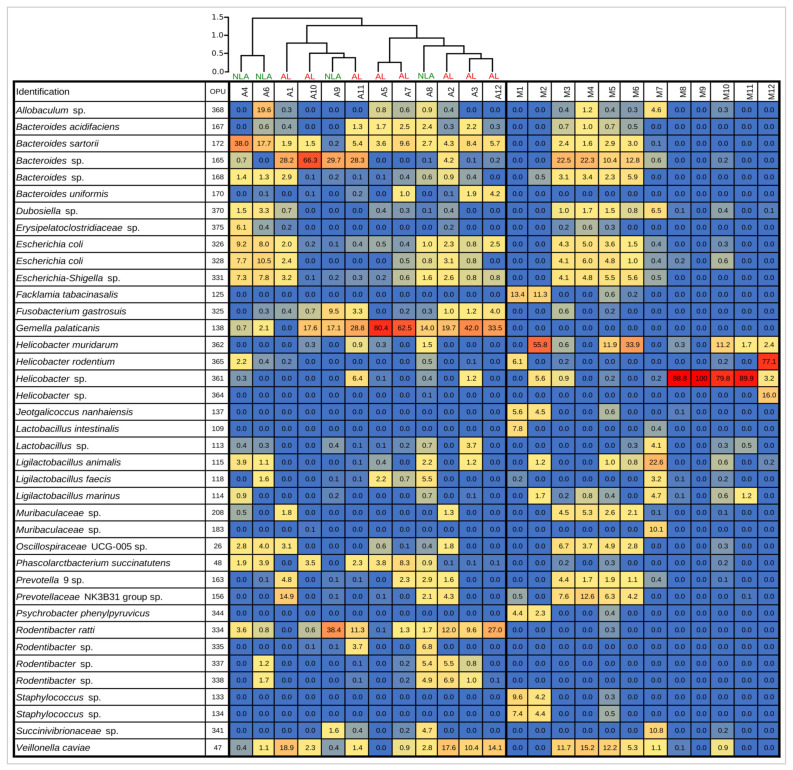
**Heat map and dendrogram analyses**. Heat map of relative abundances (%) of top OPUs per sample up to 5%. The inset color of blue indicates zero; yellow to increasing red values indicate above 1%. On top of experimental anastomosis samples (A), dendrogram analysis, Bray–Curtis distance method, and Ward’s clustering algorithm. M: control mucosa, A: anastomosis, NLA: non-leak anastomosis, AL: anastomotic leak.

**Figure 8 microorganisms-11-00680-f008:**
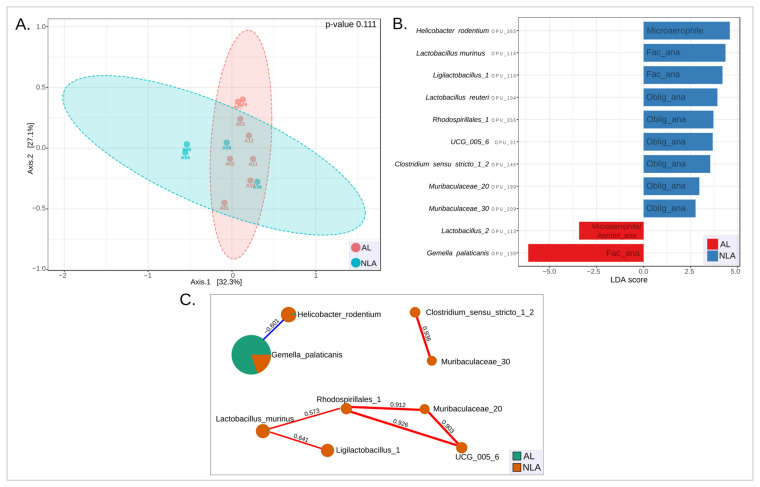
**Beta diversity, LEfSe-LDA, and correlation network analyses of NLA and AL groups.** (**A**) PCoA of NLA and AL samples using Bray–Curtis beta diversity measures. Statistics: PERMANOVA. (**B**) LEfSe-LDA score >2 for the identification of species that differ significantly in abundance between groups (raw *p*-value cutoff < 0.05, Kruskal–Wallis test). (**C**) Correlation networks (SparCC) of 11 species depicted in LEfSe-LDA. Correlation threshold > 0.3 and *p*-value threshold < 0.05 were used. The nodes (pie charts) represent species, size reflects relative abundance, with the AL group in green and the NLA group in orange. Positive and negative correlations in red and blue lines, respectively. The thickness of the line represents the strength of the correlation. Fac_ana: facultative anaerobe, Oblig_ana: obligate anaerobe, Aerotol_ana: aerotolerant anaerobe, NLA: non-leak anastomosis, AL: anastomotic leak.

## Data Availability

The data presented in this study are available on request from the corresponding author.

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
