# Peer review of "Microbiota Phenotype Promotes Anastomotic Leakage in a Model of Rats with Ischemic Colon Resection"

_microorganisms, 2023, doi:10.3390/microorganisms11030680_

Round 1

Reviewer 1 Report

The future work for the authors to do should be the evaluation of how microbiota phenotype promotes anastomotic leakage in patients with ischemic colon resection. Please discuss the significance, similarity, difference and challenges between the rat model and clinical trials to show that microbiota phenotype promotes anastomotic leakage in patients with ischemic colon resection.

Author Response

Responses to reviewers of Zamorano et al.: " Microbiota phenotype promotes anastomotic leakage in a model of rats with ischemic colon resection.".

Manuscript: microorganisms-2165074 version 2

Dear Editor,

On behalf of all the authors, I would like to thank to the Reviewers for their advice and suggestions on the revision of our manuscript "Microbiota phenotype promotes anastomotic leakage in a model of rats with ischemic colon resection" by Zamorano et al. We have responded point by point to the issues and modified the manuscript according to the recommendations of Reviewers 1 and 2.

Moreover, we want to thank the reviewers for the insightful comments and careful review of our manuscript; they have greatly increased the quality of our manuscript.

Reviewer 1

The future work for the authors to do should be the evaluation of how microbiota phenotype promotes anastomotic leakage in patients with ischemic colon resection. Please discuss the significance, similarity, difference and challenges between the rat model and clinical trials to show that microbiota phenotype promotes anastomotic leakage in patients with ischemic colon resection.

Response: We appreciate the reviewer's feedback and have attempted to address their concerns about the animal model. We agreed with the reviewer that the animal model issue should be discussed in greater depth (a text has been included in the discussion section lines 445-478).

The best animal model (1) should be able to reproduce the disease or event under study with the greatest anatomical and physiological resemblance to a human. However, it must also be widely available, inexpensive, and technically feasible to perform in a safe and repeatable manner, and the animal should be easy to keep in a suitable space.

Animal models used are usually very different from the typical clinical setting in humans. Models are generally young and healthy animals, whereas humans are typically old and fragile. The above would explain why AL is less common in animal models, so some pathophysiological mechanisms (e.g., insufficient sutures, ischemia) must be simulated for filtration to occur (2-3).

AL models have been performed in rats and mice. These animals are excellent for these purposes because they are low cost, easy to maintain and handle, and widely available. Unlike strict herbivores, these animals can digest food derived from animals, and their digestive system is less fragile, being a model of greater similarity to humans (4). Most studies have been conducted in rats with discouraging initial results, where despite models with insufficient or ischemic sutures, the filtration could not be reproduced. The authors conclude that due to the greater consistency of the stools, creating a colorectal anastomosis that presents a leak is harder (5). In other models with the use of corticosteroids, AL was evidenced, but the omentum contained it without showing an obvious filtration (6). Recently, rats with ischemic colorectal anastomosis were used, which was achieved by the devascularization of a proximal and distal centimeter of the anastomosis. A filtration rate of 50% confirmed by the autopsy of the specimens on the sixth day was achieved postoperatively, showing only one death during anesthesia (7), which makes it an excellent model to study this event, being reproducible, economical, with a high incidence of the event to be studied and safe (1).

Like the Shogan study (8), our rat model has certain limitations in terms of recapitulating the anastomotic leak in humans. Leaks are not directly visualized during surgery as most patients do not undergo reoperation. In addition, most anastomotic leaks are discovered by computed tomography (CT) scans and appear as inflammation or fluid collections adjacent to the anastomotic site. The precise definition of AL is thus debated, and contrast enema and CT scanning are known to carry significant false negative rates. Our approach examines all anastomoses in rats for evidence of dense adhesion, dehiscence, inflammation, and purulence that contrast with a healthy non-inflamed anastomosis without adhesions. We believe this assessment aligns with what might be seen clinically in patients.

  1. Chow PKH: Using Animal Models in Biomedical Research: A Primer for the Investigator. River Edge, World Scientific, 2008.
  2. Komen N, van der Wal HC, Ditzel M, Kleinrensink GJ, Jeekel H, Lange JF: Colorectal anastomotic leakage: a new experimental model. J Surg Res 2009;155:7–12.
  3. Van der Ham AC, Kort WJ, Weijma IM, van den Ingh HF, Jeekel H: Healing of ischemic colonic anastomosis: fibrin sealant does not improve wound healing. Dis Colon Rectum 1992;35:884–891.
  4. Hem A, Eide DM, Engh E, Smith A (eds): Laboratory Animal Science. Oslo, Norwegian School of Veterinary Science, 2001.
  5. Van der Ham AC, Kort WJ, Weijma IM, Jeekel H: Transient protection of incomplete colonic anastomoses with fibrin sealant: an experimental study in the rat. J Surg Res 1993;55:256– 260
  6. Henne-Bruns D, Kreischer HP, Schmiegelow P, Kremer B: Reinforcement of colon anastomoses with polyglycolic acid mesh: an experimental study. Eur Surg Res 1990;22:224– 230.
  7. Shakhsheer BA, Lee B, Guyton K, Defnet AM, Bagrodia N, Kandel JJ, Zaborina O, Hernandez SL, Alverdy J (2016) Lack of evidence for tissue hypoxia as a contributing factor in anastomotic leak following colon anastomosis and segmental devascularization in rats.Int J colorectal Dis
  8. Shogan BD, Belogortseva N, Luong PM, Zaborin A, Lax S, Bethel C, Ward M, Muldoon JP, Singer M, An G, Umanskiy K, Konda V, Shakhsheer B, Luo J, Klabbers R, Hancock LE, Gilbert J, Zaborina O, Alverdy JC. Collagen degradation and MMP9 activation by Enterococcus faecalis contribute to intestinal anastomotic leak. Sci Transl Med. 2015 7(286):286ra68.

Reviewer 2 Report

Zamorano et al. have analysed the changes in the microbiota of colon resection in a rat model. The work is sound but I want to address some concerns to the authors.

Major concerns

1)      In the methods is not clearly stated, but reading the results I can guess that the comparisons made were Anastomosis vs Control Mucosa; and Non-Leak Anastomosis vs Anastomosis Leak. Did you try Non-Leak Anastomosis vs Control Mucosa; and Anastomosis Leak vs Control mucosa? I see that micorbioa respiration yes, but in the rest? If we are naïve, we can think that Non-Leak Anastomosis would be similar to controls, since everything is healing properly, while in Asastomosis Leak more differences would be seen. For example, take Figure2C, in Chao-1, NLA seems to be higher than controls (in spite of the variability) and similar in AL vs M; in Eveness is seems that AL is lower than M, etc. I do not if you have tried it.

2)      Since only male rats have been used, in the discussion should be highlighted that limitation. The conclusions based on the results are only relevant for male rats, ideally, in female rats should be the same, but we do not know. In addition, we do not know if the proposed mechanism to explain the shift in the microbiota (oxygen influx) can be transferred to all human patients or only males. It would be nice to have this added in the discussion.

3)      In the discussion is analysed the role of G. palaticanis in terms of genes and biological functions that can be doing (lines 533-544). Actually, it is a very interesting point to know if there are changes in the microbial functions, specially the ones related to cell adhesion. Thus, it would be interesting the use of Picrust (https://huttenhower.sph.harvard.edu/picrust/) to predict the functional changes. I guess that it can be used with PacBio data and it would add relevant information to the study. 

Minor concerns

1)      In the introduction it is stated that “Although previous studies provide evidence for the hypothesis that the microbiota may have a cause role in the AL” (lines 62-63), but not references are provided. This is referring human studies or the rat models mentioned after? I think that it is relevant for the reader if the role of microbiota in AL have been studied in humans.

2)      In methods, it should be better explained how have been carried out (programs, options, etc.) the PCoA, HeatTree, LEfSe, correlations and annotation of microbial respiration. 

3)      In results, some paragraphs seems more discussion (e. g. 253-262; 358-363; etc.) than results, can you revise it?

4)      In Figure 3 should be nice to see what A samples are NLA. Based on Figure 8A, it can be seen that A04 A06 are NLA and they clustered with control mucosa, something that seems interesting.

5)      In the label of Figure 6, the explanation of panel C is missing

6)      At the end of the discussion (lines 545-548), the paragraph from the layout is still there.

Author Response

Responses to reviewers of Zamorano et al.: " Microbiota phenotype promotes anastomotic leakage in a model of rats with ischemic colon resection.".

Manuscript: microorganisms-2165074 version 2

Dear Editor,

On behalf of all the authors, I would like to thank to the Reviewers for their advice and suggestions on the revision of our manuscript "Microbiota phenotype promotes anastomotic leakage in a model of rats with ischemic colon resection" by Zamorano et al. We have responded point by point to the issues and modified the manuscript according to the recommendations of Reviewers 1 and 2.

Moreover, we want to thank the reviewers for the insightful comments and careful review of our manuscript; they have greatly increased the quality of our manuscript.

Reviewer 2

Zamorano et al. have analyzed the changes in the microbiota of colon resection in a rat model. The work is sound, but I want to address some concerns to the authors.

Major concerns

1)      In the methods is not clearly stated, but reading the results I can guess that the comparisons made were Anastomosis vs Control Mucosa; and Non-Leak Anastomosis vs Anastomosis Leak. Did you try Non-Leak Anastomosis vs Control Mucosa; and Anastomosis Leak vs Control mucosa? I see that micorbiota respiration yes, but in the rest? If we are naïve, we can think that Non-Leak Anastomosis would be similar to controls, since everything is healing properly, while in Asastomosis Leak more differences would be seen. For example, take Figure2C, in Chao-1, NLA seems to be higher than controls (in spite of the variability) and similar in AL vs M; in Eveness is seems that AL is lower than M, etc. I do not if you have tried it.

Response: We agree with reviewer's concern. In this regard, all comparisons were made at first, and no differences were found. The high variability among samples (observable in the magnitude of standard deviations) and the small sample size, could explain the above mentioned. As a result, the decision was made to include the graphs that best demonstrated the observations we wanted to highlight. Comparisons between the anastomotic leakage (AL) and non-leak anastomosis (NLA) groups were made because previous studies have found a decrease in alpha diversity when comparing individuals who have undergone anastomotic surgery.

Therefore, the reviewer's comments are important because they clarify additional observations. As a result, a sentence has been added (bold text) to the paragraph on line 245 of the original text:

“When the control (M) and experimental anastomosis (A) groups were analyzed, it was found that they both had similar alpha diversity indices, without showing significant differences. The same happens when analyzing the NLA and AL groups (Figure 2C, Table S2). Even when comparing the three groups, M, NLA and AL, no differences were observed (data not shown). Despite this, a trend towards a lower presence of estimated OPUs in Chao-1 can be observed for the AL group when compared to the NLA group. In addition to this, there is also a trend towards greater dominance and less heterogeneity in AL given the Shannon-Weiner Index. Finally, lower values for the AL group can be seen in evenness, indicating a less uniform distribution of species (OPUs). All these results show us a tendency to less diversity in the AL samples.

2)      Since only male rats have been used, in the discussion should be highlighted that limitation. The conclusions based on the results are only relevant for male rats, ideally, in female rats should be the same, but we do not know. In addition, we do not know if the proposed mechanism to explain the shift in the microbiota (oxygen influx) can be transferred to all human patients or only males. It would be nice to have this added in the discussion.

Response: We appreciate the reviewer's feedback. In this regard, we can say that the goal of this study was to add evidence to the hypothesis about the role of microbiota in AL. Other factors, such as the animal model's gender, age, diet, or comorbidities, could influence how the microbiota affects AL. However, we believe that the effect of oxygen on microbiota bacteria is a very general mechanism that can be assumed to be sufficiently independent of the host to consider sex as a determinant in this observation.

Chamorro et al. (1) recently published a paper in which they identified potential co-occurring and mutually exclusive interactions between bacteria associated with healthy and inflamed mucosa, apparently related to oxygen availability and type of respiration. This observation was made regardless of the gender of the patient.

We have added the following text to the manuscript in lines 492-497:

“Other factors, such as sex, age, diet, or the animal model's comorbidities, could influence how the microbiota affects AL but were not considered in this study. However, we believe that the effect of oxygen on microbiota bacteria is a general mechanism that is sufficiently independent of host factors to account for our findings.”

3)      In the discussion is analysed the role of G. palaticanis in terms of genes and biological functions that can be doing (lines 533-544). Actually, it is a very interesting point to know if there are changes in the microbial functions, specially the ones related to cell adhesion. Thus, it would be interesting the use of Picrust (https://huttenhower.sph.harvard.edu/picrust/) to predict the functional changes. I guess that it can be used with PacBio data and it would add relevant information to the study.

Response: While it might provide us with new information, we would like the reviewer to understand our situation.

Although the reviewer suggests using PRICRUST to predict the functions of our microbiota data, this analysis is beyond the scope of the current study. For the analysis of specific adhesion genes, PICRUST only allows us to address broader gene families (KEGG orthologs and Enzyme Classification numbers) (2) rather than the detailed species-level analysis we desired. Furthermore, because we used complete 16S gene amplicons in this study, which allow for maximum taxonomic resolution down to the species level, the detailed description of the species we focused on, particularly Gemella palaticanis, could be completed without difficulty.

If we had used short 16s sequences like Illumina, which allow for maximum taxonomic resolution down to the genus level, the results would have been the opposite. More general analyses, such as those obtained by PICRUST, can help to supplement the less detailed information obtained from taxonomy in these cases. We cannot link the study of OPUs with PICRUST because the input to PICRUST is a table generated by QIIME. It would force us to try to perform this analysis at the ASV/OTU level, for which we have no experience with PICRUST and it is not our goal.

Minor concerns

1)      In the introduction it is stated that “Although previous studies provide evidence for the hypothesis that the microbiota may have a cause role in the AL” (lines 62-63), but not references are provided. This is referring human studies, or the rat models mentioned after? I think that it is relevant for the reader if the role of microbiota in AL have been studied in humans.

Response: The reference (12) was added.

2)      In methods, it should be better explained how have been carried out (programs, options, etc.) the PCoA, HeatTree, LEfSe, correlations and annotation of microbial respiration.

Response: We agree with the reviewer's comment, so modifications have been made (bold text) in the paragraph on line 184 of the original text.

“Alpha diversity indices and rarefaction curves  were calculated using the PAST software v4.05 [22] on bacterial reads of each sample. For data analysis, OPUs harboring < 2 sequences and those occurring in a single sample were removed for beta diversity analysis and comparative analysis. Data scaling was performed by the total sum scaling (TSS) method, transforming raw reads into relative abundances by dividing the number of reads of an OPU in each sample by the total number of reads. The MicrobiomeAnalyst web-based tool [23] was used to conduct the OPU profile analyses at different taxonomic levels and the β-diversity analyses (PCoA, HeatTree, LEfSe and correlations network). The options and parameters selected can be found in the description of each figure. The annotation of microbial respiration was generated manually, using information from SILVA references [18] and papers related to the description of each species. Venn diagrams were carried out with the "InteractiVenn" web application [24]. Statistical differences between groups were performed utilizing the stats v4.1.2 R package [25]. Normality and homoscedasticity were checked. Then, parametric or non-parametric tests were performed. For post hoc pairwise comparisons, a Bonferroni or Tukey adjusted test was performed.”

3)      In results, some paragraphs seems more discussion (e. g. 253-262; 358-363; etc.) than results, can you revise it?

Response: Dear Reviewer, in this case, we have relied on the instructions for the results section that indicate:

“This section may be divided by subheadings. It should provide a concise and precise description of the experimental results, their interpretation, as well as the experimental conclusions that can be drawn”.

4)      In Figure 3 should be nice to see what A samples are NLA. Based on Figure 8A, it can be seen that A04 A06 are NLA and they clustered with control mucosa, something that seems interesting.

Response: It was added in the description of Figure 3:

Samples A4, A6, A8, A9 correspond to NLA. The remaining A samples correspond to AL.

5)      In the label of Figure 6, the explanation of panel C is missing

Response: The description in Figure 6 was modified by:

(A) Relative abundances (%) of species classified by respiration type. (B) For M and A comparations, Wilcoxon rank sum test was used. (C) For M, NLA and AL, Kruskal–Wallis and Bonferroni’s post hoc were performed. B and C graphs depict relative abundances (Log10) where each data point represents a sample, and the horizontal lines represent the means. Error bars depict standard deviations. Statistical significance: **p0.005. M: control mucosa, A: anastomosis, NLA: non-leak anastomosis, AL: anastomotic leak.

6)      At the end of the discussion (lines 545-548), the paragraph from the layout is still there.

Response: The correction was made.

References:

  1. Chamorro N, Montero DA, Gallardo P, Farfán M, Contreras M, De la Fuente M, Dubois K, Hermoso MA, Quera R, Pizarro-Guajardo M, Paredes-Sabja D, Ginard D, Rosselló-Móra R, Vidal R. Landscapes and bacterial signatures of mucosa-associated intestinal microbiota in Chilean and Spanish patients with inflammatory bowel disease. Microb Cell. 2021 Jun 18;8(9):223-238. doi: 10.15698/mic2021.09.760.
  2. Douglas, G.M., Maffei, V.J., Zaneveld, J.R. et al. PICRUSt2 for prediction of metagenome functions. Nat Biotechnol 38, 685–688 (2020). https://doi.org/10.1038/s41587-020-0548-6

Round 2

Reviewer 1 Report

The manuscript has been significantly improved and suitable for publishing.

Reviewer 2 Report

I thank the authors for addressing to all the concerns.